# COMPOSITIONAL PROMPT TUNING WITH MOTION CUES FOR OPEN-VOCABULARY VIDEO RELATION DETECTION

**Kaifeng Gao**[1], **Long Chen**[2]*, **Hanwang Zhang**[3], **Jun Xiao**[1], **Qianru Sun**[4]
[1]Zhejiang University, [2]The Hong Kong University of Science and Technology
[3]Nanyang Technological University, [4]Singapore Management University
[1]{kite_phone,junx}@zju.edu.cn, [2]zjuchenlong@gmail.com
[3]hanwangzhang@ntu.edu.sg, [4]qianrusun@smu.edu.sg

## ABSTRACT

Prompt tuning with large-scale pretrained vision-language models empowers open-vocabulary predictions trained on limited base categories, *e.g.*, object classification and detection. In this paper, we propose *compositional* prompt tuning with *motion cues*: an extended prompt tuning paradigm for compositional predictions of video data. In particular, we present Relation Prompt (RePro) for Open-vocabulary Video Visual Relation Detection (Open-VidVRD), where conventional prompt tuning is easily biased to certain subject-object combinations and motion patterns. To this end, RePro addresses the two technical challenges of Open-VidVRD: 1) the prompt tokens should respect the two different semantic roles of subject and object, and 2) the tuning should account for the diverse spatio-temporal motion patterns of the subject-object compositions. Without bells and whistles, our RePro achieves a new state-of-the-art performance on two Vid-VRD benchmarks of not only the base training object and predicate categories, but also the unseen ones. Extensive ablations also demonstrate the effectiveness of the proposed compositional and multi-mode design of prompts. Code is available at https://github.com/Dawn-LX/OpenVoc-VidVRD.

## 1 INTRODUCTION

Video visual relation detection (VidVRD) aims to detect the visual relationships between object tracklets in videos as ⟨subject, predicate, object⟩ triplets (Shang et al., 2017; Chen et al., 2021; 2023; Gao et al., 2021; 2022), *e.g.*, dog-towards-child shown in Figure 1. Compared to its counterpart in still images (Chen et al., 2019; Li et al., 2022b;c;d;e), due to the extra temporal axis, there are usually multiple relationships with different temporal scales, and a subject-object pair can have several predicates with ambiguous boundaries. For example, as shown in Figure 1, the action feed of child to dog co-occurs with several other predicates (*e.g.*, away, towards).

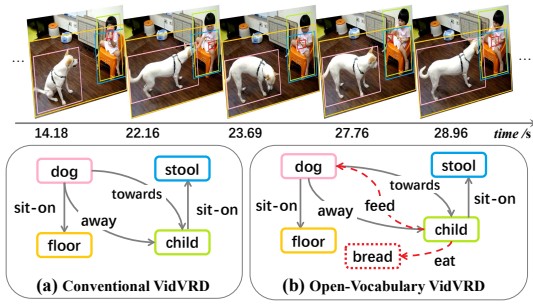

Figure 1: Examples of VidVRD. The relation graphs are w.r.t the whole video clip. Dashed lines denote unseen *new* categories in the training data.

This characteristic makes VidVRD have more plentiful and diverse relations between objects than its image counterpart. As a result, it is impractical to collect sufficient annotations for all categories for VidVRD. Therefore, to make VidVRD practical, we should know how to generalize the model, trained on limited annotations, to *new* object and predicate classes *unseen* in training data.

---

*Long Chen is the corresponding author. Part of the work was done when Kaifeng Gao served as a visiting Ph.D. student at Singapore Management University.

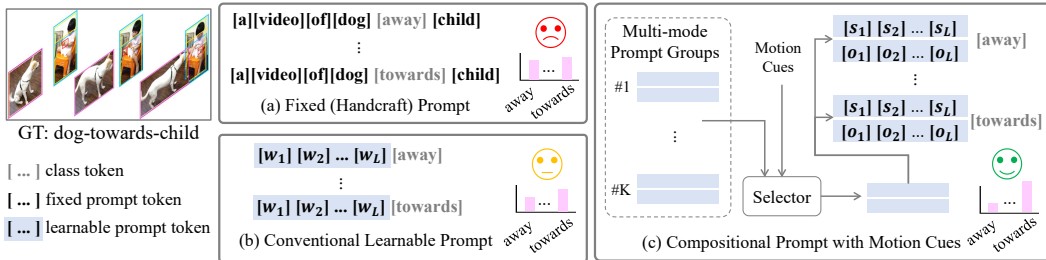

Figure 2: Comparisons of different prompt tuning methods for Open-VidVRD.

To this end, we propose a new task: *Open-vocabulary* VidVRD (Open-VidVRD). In particular, "open" does not only mean unseen relationship combinations, *e.g.*, `dog-sit-on-floor`, but also unseen objects and predicates, *e.g.*, `bread` and `feed`, as shown in Figure 1. Recent works on such generalization only focus on the unseen combinations (Chen et al., 2021; Shang et al., 2021) in Vid-VRD, or zero-shot transfer among semantically related objects in zero-shot object detection (Huang et al., 2022), *e.g.*, the seen `dog` class can help to recognize the unseen `wolf`. However, they fail to generalize to the categories totally unrelated to the limited seen ones, where the transfer gap is unbridgeable, *e.g.*, `bread` in testing has no visual similarity with `dog` and `child` in training.

Thanks to the encyclopedic knowledge acquired by large vision-language models (VLMs) pre-trained on big data (Radford et al., 2021; Li et al., 2022a), we can achieve open-vocabulary relation detection with only training data of limited base categories. To bridge the gap between the pre-trained and downstream tasks without extra fine-tuning the whole VLM model, a trending technique named **prompt tuning** is widely adopted (Liu et al., 2021; Jin et al., 2022; Zhou et al., 2022b). For example, we can achieve zero-shot relation classification for the tracklets pair in Figure 2. We first crop the object tracklet regions in the video, and feed them into the visual encoder of VLM to obtain corresponding visual embeddings. Then we use a simple prompt like "a video of [CLASS]", feed it to the VLM's text encoder to obtain the text embedding, and classify the object based on the similarities between visual and text embeddings. Based on the tracklet classification results, for the example of the pair `dog` and `child`, we can craft a prompt like "a video of dog [CLASS] child", as shown in Figure 2(a), and similarly classify their predicates based on the predicate text embeddings. Furthermore, we can replace the fixed prompt tokens with learnable continuous tokens, as shown in Figure 2(b), known as prompt representation learning, which has been widely applied to open-vocabulary object detection (Gu et al., 2021; Du et al., 2022; Ma et al., 2022).

Learning the prompt representation is actually introducing some priors for describing the context of target classes, and it excludes some impossible classes with the constraint of the context. However, the prompt representations (either handcrafted or learned) in above approaches are monotonous and static, and learning the prompt sometimes might break the "open" knowledge due to overfitting to the base category training data. Modeling the prompt representation for video visual relations has some specific characteristics that need to be considered:

- **Compositional**: The prompt context for predicates is highly related to the semantic roles of subject and object. A holistic prompt representation might be sub-optimal for predicates. For example, as shown in Figure 1, even the same predicate (`sit-on`) in different relation triplets (`dog-sit-on-floor` and `child-sit-on-stool`) have totally different visual context.

- **Motion-related**: Predicates with different motion patterns naturally should be prompted with different context tokens. The naive prompt representation fails to consider the spatio-temporal motion cues of tracklet pairs. For example, the predicate `towards` shown in Figure 1 can be prompted as "a relation of [CLASS], moving closer". In contrast, `eat` and `sit-on` can be prompted as "a relation of [CLASS], relative static".

In this paper, we propose a compositional and motion-based **Re**lation **Pro**mpt learning framework: RePro, as shown in Figure 2(c). To deal with the **compositional** characteristic of visual relations, we set compositional prompt representations specified with subject and object respectively. With this design, we can model the prompt context w.r.t. semantic roles (*i.e.*, `subject` or `object`). For example, a possible prompt can be "sth. doing [CLASS]" for the subject and "sth. being [CLASS]" for the object. To consider the **motion-related** characteristic of predicate contexts, we design multi-mode prompt groups, where each group (*i.e.*, each mode) is assigned with a certain motion pattern, and has its own compositional prompts for the subject and object. During the implementation, we

select a proper group according to the motion cues (patterns) in the subject-object tracklet pairs (cf. Sec. 3.3). Compared to some prompt tuning works which focus on category-based context (Zhou et al., 2022b) or instance-conditioned context (Zhou et al., 2022a; Ni et al., 2022), our motion-cue-based grouping has better cross-category generalization ability, and can avoid the over-fitting to base categories. We evaluate our RePro on the VidVRD (Shang et al., 2017) and VidOR (Shang et al., 2019) benchmarks. Our experiment results show that RePro trained with only the samples of *base* relation categories has a good generalizability to detect *novel* relations, and achieves the new state-of-the-art. For example, it outperforms the top-performing method, *i.e.*, VidVRD-II (Shang et al., 2021), by 2.54% and 3.91% absolute mAP for SGDet and SGCls settings, respectively.

Our contributions in this paper are thus three-fold. 1) A new open-vocabulary setting for video visual relation detection task, *i.e.*, Open-VidVRD. 2) A compositional prompt representation learning method that models the prompt contexts for the subject and object separately. 3) A motion-cue-based multi-mode prompt groups that achieve a strong generalization ability.

## 2    RELATED WORK

**Video Visual Relation Detection (VidVRD)** was defined in Shang et al. (2017; 2019) together with the proposals of the VidVRD and VidOR benchmarks. The task aims to spatio-temporally localize visual relations between object tracklets. Existing methods mainly focus on modeling better visual or spatio-temporal contexts (Qian et al., 2019; Shang et al., 2021; Cong et al., 2021), and detecting visual relations with more granularity either by sliding windows (Liu et al., 2020) or temporal grounding (Gao et al., 2022). They mainly worked on the pre-defined (closed) sets of object and predicate categories. In contrast, our work is the first one to study the open-vocabulary VidVRD setting, where some object and predicate categories are unseen in the training set.

**Zero-Shot Setting in Image and Video VRD**. Existing VRD works, either in image domain (Tang et al., 2020; Kan et al., 2021) or video domain (Shang et al., 2021), only achieve zero-shot transfer on the unseen triplet combinations, where the objects and predicates are seen in the training set. They ignore the model's generalization ability to unseen object/predicate categories. There is one concurrent work (He et al., 2022) proposes the open-vocabulary setting in image VRD. However, they put the main emphasis on unseen object categories. Different from them, RePro generalizes the model to recognize both object and predicate categories totally unrelated to the seen training ones.

**Prompt Tuning for Open-vocabulary Visual Recognition**. Prompt tuning (Liu et al., 2021) has been widely adopted in both image (*e.g.*, open-vocabulary object detection (OV-Det) (Gu et al., 2021; Du et al., 2022; Ma et al., 2022)) and video (*e.g.*, zero-shot video action recognition (Lin et al., 2022; Ni et al., 2022; Ju et al., 2022; Nag et al., 2022)) domains. For OV-Det, recent works mainly focus on knowledge distillation from VLM and simply using handcrafted prompt (Gu et al., 2021; Ma et al., 2022), or focus on prompt representation learning for object regions (Du et al., 2022; Feng et al., 2022). For video action recognition, existing works mainly use fixed prompt (Nag et al., 2022), conventional learnable prompt (Ju et al., 2022), or prompt conditioned on the input video contexts (Ni et al., 2022), and they all focus on the cross-frame attention or feature interaction. In contrast, our RePro learns the compositional prompt by leveraging the motion cues of subject-object pairs, and has better cross-category generalization ability to detect *novel* visual relations in videos.

## 3    METHOD

To build the Open-VidVRD setting, we first divide the categories of a dataset into *base* and *novel* splits. Specifically, we denote $\mathcal{C}_b^O$ and $\mathcal{C}_n^O$ as the sets of *base* and *novel* object categories, respectively. We use $\mathcal{C}_b^P$ and $\mathcal{C}_n^P$ to denote the sets of *base* and *novel* predicate categories, respectively. In the training stage, we use all visual relation triplet samples from $\mathcal{C}_b^O \times \mathcal{C}_b^P \times \mathcal{C}_b^O$. In the testing stage, we evaluate the model with the triplets sampled from all categories, *i.e.*, $\mathcal{C}_b^O \cup \mathcal{C}_n^O$ and $\mathcal{C}_b^P \cup \mathcal{C}_n^P$.

Based on this new setting, we first briefly introduce the preliminaries for open-vocabulary classification with pre-trained VLMs (Sec. 3.1). Then, we introduce the proposed Open-VidVRD method RePro, as illustrated in Figure 3, in which we first extend open-vocabulary object detection methods (Gu et al., 2021; Du et al., 2022) to open-vocabulary tracklet detection (Sec. 3.2), and then perform open-vocabulary relation classification for each tracklet pair (Sec. 3.3).

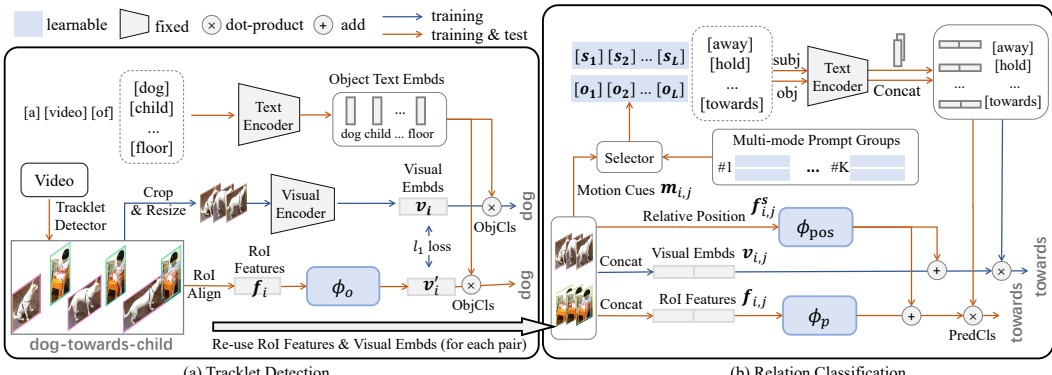

Figure 3: The overall pipeline of RePro. The visual embeddings ($\boldsymbol{v}_i, \boldsymbol{v}_{i,j}$) are only used at training time, in which we train two V2L project modules (*i.e.*, $\phi_o$ and $\phi_p$) to transfer the knowledge from the pre-trained VLM. At test time, only RoI features ($\boldsymbol{f}_i, \boldsymbol{f}_{i,j}$) and position features ($\boldsymbol{f}_{i,j}^s$) are used. For tracklet detection (a), the knowledge is transferred by aligning $\boldsymbol{v}_i'$ to $\boldsymbol{v}_i$. For relation classification (b), the knowledge is transferred by the prompt representations learned in the supervision of $\boldsymbol{v}_{i,j}$.

## 3.1 PRELIMINARIES: OPEN-VOCABULARY CLASSIFICATION WITH PROMPT

**Fixed Prompt**. Pre-trained VLMs have a strong open-vocabulary classification ability (Li et al., 2022a). They first extract text embeddings for all categories by feeding handcrafted prompt (*e.g.*, "a video of [CLASS]") into the text encoder of VLM, where [CLASS] can be replaced with the class name of an arbitrary object or predicate. Their output text embedding $\boldsymbol{t}_c \in \mathbb{R}^d$ for each class $c$ is

$$\boldsymbol{t}_c = \text{VLM}_{\text{txt}}(\boldsymbol{W}_c), \ \boldsymbol{W}_c = [\boldsymbol{w}_1, \ldots, \boldsymbol{w}_L, \tilde{\boldsymbol{w}}_c], \forall c \in \mathcal{C}_b^O \cup \mathcal{C}_n^O \text{ or } \mathcal{C}_b^P \cup \mathcal{C}_n^P, \tag{1}$$

where $\boldsymbol{W}_c$ is the prompt representation with $L$ context token vectors and the class token vector $\tilde{\boldsymbol{w}}_c$. Then, for each object tracklet with cropped video region, the corresponding visual embedding can be extracted by the visual encoder of VLM, denoted as $\boldsymbol{v}_i' \in \mathbb{R}^d$. Similarly, the visual embedding of tracklet pair $(i, j)$ can also be extracted (*e.g.*, based on the union region), denoted as $\boldsymbol{v}_{i,j}$. Therefore, the $i$-th region (generally denoted as $\boldsymbol{v}_i$) can be classified by the cosine similarities w.r.t $\{\boldsymbol{t}_c\}$:

$$\hat{c}_i = \arg\max_c \cos(\boldsymbol{v}_i, \boldsymbol{t}_c), \ \forall c \in \mathcal{C}_b^O \cup \mathcal{C}_n^O \text{ or } \mathcal{C}_b^P \cup \mathcal{C}_n^P, \text{ where } \cos(\boldsymbol{x}, \boldsymbol{y}) = \boldsymbol{x}^{\text{T}}\boldsymbol{y}/(\|\boldsymbol{x}\|\|\boldsymbol{y}\|). \tag{2}$$

**Learnable Prompt**. Manually tuning the words in the prompt requires domain expertise and is time-consuming or not robust (Radford et al., 2021). The substitute method is to learn the prompt representations from the training data (Zhou et al., 2022b;a). Specifically, the context vector $\boldsymbol{w}_i$ in $\boldsymbol{W}_c$ can be set as a learnable vector while $\tilde{\boldsymbol{w}}_c$ is kept as fixed. In the training stage, samples and $\{\tilde{\boldsymbol{w}}_c\}$ are from base categories. In the testing stage, the $L$ learned vectors in each $\boldsymbol{W}_c$ are fixed and then the model performs classification in the same way as Eq. (2).

## 3.2 OPEN-VOCABULARY OBJECT TRACKLET DETECTION

**Tracklet Proposal Generation**. Given a video, we first detect all the class-agnostic object tracklets using a pre-trained tracklet detector, denote as $\mathcal{T} = \{T_i\}_{i=1}^N$, as shown in Figure 3(a). Specifically, each tracklet $T_i$ is characterized with a bounding box sequence and the corresponding RoI Aligned (He et al., 2017) visual feature, To reduce the computational overhead, we average the RoI features of all bounding boxes (*i.e.*, along the temporal axis of the tracklet) following (Shang et al., 2021), and denote it as $\boldsymbol{f}_i \in \mathbb{R}^{2048}$.

**Tracklet Classification**. Instead of directly classifying object tracklets using VLM as Eq. (2), we train a visual-to-language (V2L) projection module $\phi_o(\cdot)$ to further utilize the annotations of base classes. In particular, $\phi_o(\cdot)$ maps the RoI Aligned feature $\boldsymbol{f}_i$ of each tracklet to the same semantic space $\mathbb{R}^d$, *i.e.*, $\boldsymbol{v}_i' = \phi_o(\boldsymbol{f}_i)$. Let $\boldsymbol{t}_c^o$ be the text embedding of object class $c \in \mathcal{C}_b^O$. The probability of tracklet $T_i$ being classified as class $c$ can be calculated as

$$p_i(c) = \frac{\exp(\cos(\boldsymbol{v}_i', \boldsymbol{t}_c^o)/\tau)}{\sum_{c' \in \mathcal{C}_b^O} \exp(\cos(\boldsymbol{v}_i', \boldsymbol{t}_{c'}^o)/\tau)}, \forall c \in \mathcal{C}_b^O, \tag{3}$$

where $\tau$ is a temperature parameter for softmax.

**Training Objectives**. To train the object tracklet classification module, we assign base category labels to detected tracklets according to the IoU w.r.t ground-truth tracklets. We call those tracklets with assigned labels as positive tracklets, otherwise negative tracklets. Note that two cases can be recognized as negative tracklets: 1) the content is background; and 2) the content contains a novel object category. For these negative tracklets, we follow the loss used by Du et al. (2022) that forces the prediction (from any negative tracklet) on each base class to be $1/|\mathcal{C}_b^O|$, *i.e.*, unlike any base category. Therefore, the classification loss for positive and negative tracklets can be calculated as:

$$\mathcal{L}_{\text{cls-pos}} = -\frac{1}{|\mathcal{T}_p|} \sum_{T_i \in \mathcal{T}_p} \sum_{c \in \mathcal{C}_b^O} \mathbb{1}_{\{c=c_i^*\}} \log p_i(c), \quad \mathcal{L}_{\text{cls-neg}} = -\frac{1}{|\mathcal{T}_n|} \sum_{T_i \in \mathcal{T}_n} \sum_{c \in \mathcal{C}_b^O} \frac{1}{|\mathcal{C}_b^O|} \log p_i(c), \quad (4)$$

where $\mathcal{T}_p$ and $\mathcal{T}_n$ are the sets of positive and negative tracklets, respectively (*i.e.*, $\mathcal{T}_p \cup \mathcal{T}_n = \mathcal{T}$), and $c_i^*$ is the ground-truth label for the $i$-th positive tracklet. We empirically found (in Sec. 3.2) that using the above negative tracklet loss works better than using the loss with a unique "background" class (Zareian et al., 2021; Gu et al., 2021). Besides, following (Gu et al., 2021), we distill the knowledge from a pre-trained visual encoder to $\phi_o(\cdot)$ by aligning $\boldsymbol{v}_i'$ to $\boldsymbol{v}_i$ using $l_1$ loss, *i.e.*,

$$\mathcal{L}_{\text{distill}} = (1/N)\textstyle\sum_{i=1}^{N}\|\boldsymbol{v}_i' - \boldsymbol{v}_i\|_1 \qquad (5)$$

Therefore, the overall loss for object tracklet classification is $\mathcal{L}_{\text{cls}} = \mathcal{L}_{\text{cls-pos}} + \mathcal{L}_{\text{cls-neg}} + \lambda\mathcal{L}_{\text{distill}}$, where $\lambda$ is a hyper-parameter to weight the classification and distillation.

## 3.3 OPEN-VOCABULARY VISUAL RELATION CLASSIFICATION

Based on the classified object tracklets, we perform open-vocabulary relation classification for each tracklet pair, as shown in Figure 3(b). First, we learn the prompt representations based on the pre-extracted visual embeddings of tracklet pairs, for which we introduce the *compositional prompt representations* and the *motion-based prompt groups*. Second, we utilize the pre-extracted RoI Aligned features to train a visual-to-language (V2L) projection module based on the learned prompt representations. Finally, for testing, we extract all predicate text embeddings and classify the predicates of each tracklet pair by using the RoI Aligned features and the trained V2L projection module.

**Compositional Prompt Representations**. The compositional prompt consists learnable prompt representations $\boldsymbol{S}_c$ and $\boldsymbol{O}_c$ (of predicate class $c$) for subject and object, respectively:

$$\boldsymbol{S}_c = [\boldsymbol{s}_1, \ldots, \boldsymbol{s}_L, \tilde{\boldsymbol{w}}_c], \quad \boldsymbol{O}_c = [\boldsymbol{o}_1, \ldots, \boldsymbol{o}_L, \tilde{\boldsymbol{w}}_c], \qquad (6)$$

where $\boldsymbol{s}_i$ and $\boldsymbol{o}_i$ are the learnable context vectors and $\tilde{\boldsymbol{w}}_c$ is the fixed class token for predicate $c$ (for $c \in \mathcal{C}_b^P$ in training phase and for all $c$ in testing phase). Then, the predicate text embedding $\boldsymbol{t}_c^p$ is generated by concatenating the two outputs of VLM given two prompts (respectively) as inputs, *i.e.*,

$$\boldsymbol{t}_c^p = [\text{VLM}_{\text{txt}}(\boldsymbol{S}_c), \text{VLM}_{\text{txt}}(\boldsymbol{O}_c)], \text{ and } \boldsymbol{t}_c^p \in \mathbb{R}^{2d}. \qquad (7)$$

**Motion-based Prompt Groups**. We vary the prompt contexts based on the motion cues, *i.e.*, the relative spatio-temporal motion patterns, between each pair of subject and object. In specific, we take the generalized IoU Rezatofighi et al. (2019) (*i.e.*, GIoU) as the metric to calculate the motion patterns. For each tracklet pair $\langle T_i, T_j \rangle$, we use a vector to represent a motion pattern:

$$\boldsymbol{m}_{i,j} = \text{sign}([G_{i,j}^s - \gamma, G_{i,j}^e - \gamma, G_{i,j}^e - G_{i,j}^s]), \text{ and } \boldsymbol{m}_{i,j} \in \{+, -\}^3, \qquad (8)$$

where $G_{i,j}^s, G_{i,j}^e$ are the GIoU between subject-object for the start and end bounding boxes of their temporal intersection, respectively, and $\gamma$ is a threshold for GIoU. This definition considers two perspectives: 1) whether the two tracklets are near or far (*i.e.*, the first two terms of Eq. (8)), and 2) whether they move toward or away to each other (*i.e.*, the third term of Eq. (8). Overall, we have 6 motion patterns (cf. Sec. A.2 for more details) and build 6 prompt groups correspondingly. Each group consists its own compositional prompt representations $\boldsymbol{S}_c$ and $\boldsymbol{O}_c$ as defined in Eq. (6).

It's worth noting that we aim to build a framework for learning motion-based multi-mode prompts. The used GIoU-based approach (in our framework) is a simple and intuitive way to calculate motion cues. This approach is not perfect, *e.g.*, it is poor to capture the motion pattern of tracklets moving back and forth. We leave other fancier (motion capturing) approaches as future work.

**Training Objectives**. Based on the above definition, we train the prompt representations with visual embeddings and relative position features. For simplicity, we show the training process of a

single group (and in the end, we derive the final loss by averaging the losses across all groups). For each tracklet pair $\langle T_i, T_j \rangle$, we first calculate its motion cue, select the corresponding prompt group, and extract the class text embeddings $\boldsymbol{t}_c^p$ for each predicate class $c \in \mathcal{C}_b^P$. Then, we take the pre-extracted visual embeddings $\boldsymbol{v}_i$ and $\boldsymbol{v}_j$, and concatenate them as the pair's visual embedding $\boldsymbol{v}_{i,j} = [\boldsymbol{v}_i, \boldsymbol{v}_j] \in \mathbb{R}^{2d}$. Following Shang et al. (2021), we additionally compute the relative position feature between bounding boxes of $T_i$ and $T_j$, denoted as $\boldsymbol{f}_{i,j}^s \in \mathbb{R}^{12}$ (cf. Sec. A.2 in the Appendix for more details). The predicted probability of predicate class $c$ in this tracklet pair is thus

$$p_{i,j}(c) = \text{Sigmoid}(\cos(\boldsymbol{v}_{i,j} + \phi_{\text{pos}}(\boldsymbol{f}_{i,j}^s), \boldsymbol{t}_c^p)), \ \forall c \in \mathcal{C}_b^P. \tag{9}$$

where $\phi_{\text{pos}}$ projects $\boldsymbol{f}_{i,j}^s$ to the same dimension as $\boldsymbol{v}_{i,j}$. Going through all *base* classes (in $\mathcal{C}_b^P$), the probability vector of tracklet pair $\langle T_i, T_j \rangle$ is generated and can be denoted as $\boldsymbol{p}_{i,j}$, *i.e.*, each dimension is calculated by Eq. (9). In the training time, we assign the predicate labels according to the IoU for each tracklet pair w.r.t the ground-truth tracklet pair by following Shang et al. (2021). We denote the sets of positive and negative tracklet pairs as $\mathcal{P}_p$ and $\mathcal{P}_n$, respectively. Due to the multi-label setting of VidVRD, we use binary cross-entropy loss for relation classification. The ground truth for positive tracklet pair in $\mathcal{P}_p$ is a binary vector of dimension $|\mathcal{C}_b^P|$, denoted as $\boldsymbol{p}_{i,j}^*$. For those negative tracklet pairs in $\mathcal{P}_n$, we optimize the probability of each *base* class to zero, *i.e.*, the ground truth is an all-zero vector. The classification loss is thus calculated as

$$\mathcal{L}_{\text{pred-cls}} = (1/|\mathcal{P}_p|)\sum_{(T_i,T_j)\in\mathcal{P}_p}\text{BCE}(\boldsymbol{p}_{i,j}, \boldsymbol{p}_{i,j}^*) + (1/|\mathcal{P}_n|)\sum_{(T_i,T_j)\in\mathcal{P}_n}\text{BCE}(\boldsymbol{p}_{i,j}, \boldsymbol{0}). \tag{10}$$

**Training V2L Projection Module**. Once the prompt representations are learned, we train a visual-to-language (V2L) projection module to use RoI Aligned features $\{\boldsymbol{f}_i\}$ as training data, and get rid of VLM's visual encoder at inference time. Given the learned prompt representations, we pre-extract the predicate class text embeddings (denoted as $\{\tilde{\boldsymbol{t}}_c^p\}$) for each prompt group and fix them. Formally, for each tracklet pair $\langle T_i, T_j \rangle$, we concatenate their RoI features as $\boldsymbol{f}_{i,j} = [\boldsymbol{f}_i, \boldsymbol{f}_j] \in \mathbb{R}^{4096}$. Then, we use a V2L projection module $\phi_p$ to project it to the same dimension as text embeddings. Similar to Eq. (9), the probability of predicate class $c$ is predicted as

$$p_{i,j}(c) = \text{Sigmoid}(\cos(\phi_p(\boldsymbol{f}_{i,j}) + \phi_{\text{pos}}(\boldsymbol{f}_{i,j}^s), \tilde{\boldsymbol{t}}_c^p)), \ \forall c \in \mathcal{C}_b^P. \tag{11}$$

where $\phi_{\text{pos}}$ is the learned spatio-temporal projection layer and is fixed. Then, we apply the same loss as defined in Eq. (10), and compute the final total loss by averaging across all groups.

**Discussions**. Intuitively, we can train the prompt representations together with the V2L projection module $\phi_p$, and use the $l_1$ loss to align $\phi_p(\boldsymbol{f}_{i,j})$ to $\boldsymbol{v}_{i,j}$, *i.e.*, distill the knowledge from the pre-trained visual encoder to the V2L module. We name this variant as RePro$^\dagger$. We justify that our RePro works better than RePro$^\dagger$ due to: 1) directly using the teacher (*i.e.*, $\boldsymbol{v}_{i,j}$) to train the prompt is intuitively better than using student (*i.e.*, projected visual embedding), and 2) the distillation makes the V2L module focus too much on the static visual alignment, rather than the dynamic relation information learned in the prompt. In experiments, we empirically show the superiority of RePro over RePro$^\dagger$.

## 4 EXPERIMENTS

### 4.1 DATASETS AND EVALUATION METRICS

**Datasets**. We evaluated our method on the VidVRD (Shang et al., 2017) and VidOR (Shang et al., 2019) benchmarks: 1) VidVRD consists of 1,000 videos, and covers 35 object categories and 132 predicate categories. We used official splits: 800 videos for training and 200 videos for testing. 2) VidOR consists of 10,000 videos, which covers 80 object categories and 50 predicate categories. We used official splits: 7,000 videos for training, 835 videos for validation, and 2,165 videos for testing. Since the annotations of VidOR-test are not released, we only evaluated models on validation set.

**Evaluation Settings**. To build the open-VidVRD setting, we manually split *base* and *novel* categories by selecting the common object and predicate categories as the *base* split, and selecting the rare ones as the *novel* split. The detailed splits are given in Sec. A.8 of the Appendix. We trained the model on the triplet samples of both *base* object and predicate categories in the training set. During testing, we evaluated the model on two settings: 1) **Novel-split**: triplet samples with all object categories and *novel* predicate categories, and 2) **All-splits**: triplet samples with all object and predicate categories, in the testing set of VidVRD (or the validation set of VidOR).

Table 1: Performance (%) of tracklet detection on objects with novel categories.

| Methods | | Distillation | BG-Embd | VidVRD-test | | VidOR-val | |
|---|---|---|---|---|---|---|---|
| | | | | R@5 | R@10 | R@5 | R@10 |
| ALPro (Li et al., 2022a) | | - | - | 41.38 | 53.81 | 34.26 | 41.72 |
| RePro | #1 | × | × | 2.17 | 3.71 | 2.33 | 3.48 |
| | #2 | × | ✓ | 32.43 | 33.36 | 7.58 | 12.37 |
| | #3 | ✓ | ✓ | 43.84 | **53.00** | 12.61 | 16.85 |
| | #4 | ✓ | × | **46.34** | 50.42 | **31.62** | **37.08** |

**Metrics**. We follow three standard evaluation tasks in scene graph generation (Zellers et al., 2018): scene graph detection (SGDet), scene graph classification (SGCls), and predicate classification (PredCls). We apply these metrics to VidVRD: a detected triplet is considered to be correct if there is the same triplet tagged in the ground truth, and both subject and object tracklets have a sufficient volume IoU (*e.g.*, 0.5) with the ground truth. Following the standard setting (Shang et al., 2017), we use mAP and Recall@K (R@K, K=50,100) as evaluation metrics.

## 4.2 IMPLEMENTATION DETAILS

**Tracklet Detector & Pre-trained VLM**. We used the Faster-RCNN (Ren et al., 2015)-based VinVL model (Zhang et al., 2021) to detect frame-level object bounding boxes and extracted corresponding RoI Alinged features, and then adopted Seq-NMS (Han et al., 2016) to generate class-agnostic object tracklets. The VinVL model was trained on out-of-domain image data without seeing any VidVRD data. For pre-trained VLM, we used ALPro (Li et al., 2022a), which was pre-trained on a wide range of video-language data, and learned the fine-grained visual region to text entity alignment.

**Relation Detection Details**. Following the popular segment-based methods (Qian et al., 2019; Shang et al., 2017; 2021), we first detected visual relations in short video segments, and then adopted greedy relation association algorithm (Shang et al., 2017) to merge the same relation triplets. The detailed hyperparameter settings are left in Sec. A.3 of the Appendix.

## 4.3 EVALUATE OPEN-VOCABULARY OBJECT TRACKLET DETECTION

We evaluated the tracklet detection part of RePro on *novel* object categories, as shown in Table 1.

**Comparison to ALPro**. A straightforward baseline to achieve open-vocabulary tracklet detection is directly applying the pre-trained VLM (ALPro in our case) by inputting the tracklet regions into its visual encoder to perform classification, as in Eq. (2). However, this has a significant computational overhead due to the heavy pipeline of ALPro's visual encoder. In contrast, our RePro requires much less computational cost, since we only use one projection layer (*i.e.*, $\phi_o$). We thus compare our RePro with the above ALPro baseline. The results in row **#4** show that RePro can achieve comparable performances on both datasets, with the projection layer $\phi_o$.

**Negative Tracklet Classification**. How to model the negative sample is a key challenge as widely discussed in many open-vocabulary object detection works. There are usually two approaches: 1) using a unique background embedding (BG-Embd) in addition to the class text embeddings (Zareian et al., 2021; Gu et al., 2021), and 2) only using the class text embeddings, and computing the loss of negative sample as $\mathcal{L}_{\text{cls-neg}}$ in Eq. (4) (Gu et al., 2021). By comparing rows **#3** and **#4** of Table 1, we find that without using background embedding (*i.e.*, as $\mathcal{L}_{\text{cls-neg}}$) achieves better recall, and outperforms the other by a large margin, especially on the more challenging VidOR benchmark. This is because the tracklets recognized as negative may be due to the fact that they contain novel objects (rather than backgrounds), and aligning their embeddings (*i.e.*, different novel class embeddings) to a unique background embedding hurts the model's recognition ability on novel objects.

**Distillation**. We verified the effectiveness of visual distillation (*i.e.*, Eq. (5)) by comparing rows **#2** and **#3** of Table 1. Obviously, the distillation helps RePro improve the detection recall by a large margin, especially for the more challenging VidOR benchmark. For row **#1**, we can observe that computing the negative tracklet classification loss as $\mathcal{L}_{\text{cls-neg}}$ without distillation has extremely low performance. This is because forcing the classification probability of negative tracklet to be $1/|\mathcal{C}_b|$ (*i.e.*, by $\mathcal{L}_{\text{cls-neg}}$) and without the guidance from the teacher (*i.e.*, without distillation) make the model has poor generalize ability to novel categories.

Table 2: Performance (%) comparision to conventional methods on VidVRD-test. Relation Tagging (RelTag) only considers the precision of relation triplets and ignores the localization of tracklets.

| Methods | Training Data | SGDet | | | RelTag | | |
|---|---|---|---|---|---|---|---|
| | | mAP | R@50 | R@100 | P@1 | P@5 | P@10 |
| Su et al. (2020) | base+novel | 19.03 | 9.53 | 10.38 | 57.50 | 41.40 | 29.45 |
| Liu et al. (2020) | base+novel | 18.38 | 11.21 | 13.69 | 60.00 | 43.10 | 32.24 |
| Li et al. (2021) | base+novel | **22.97** | 12.40 | 14.46 | **68.83** | **49.87** | **35.57** |
| Gao et al. (2022) | base+novel | 17.67 | 9.63 | 11.29 | 56.00 | 43.80 | **32.85** |
| RePro (Ours) | base | 21.33 | **12.92** | **15.94** | 59.00 | 41.09 | 28.87 |
| RePro (Ours) | base+novel | **25.55** | **13.83** | **17.33** | 62.50 | 45.80 | 32.05 |

Table 3: Performance (%) comparision of Open-VidVRD methods on VidVRD-test.

| Split | Methods | SGDet | | | SGCls | | | PredCls | | |
|---|---|---|---|---|---|---|---|---|---|---|
| | | mAP | R@50 | R@100 | mAP | R@50 | R@100 | mAP | R@50 | R@100 |
| Novel | ALPro | 1.05 | 3.14 | 4.62 | 3.69 | 7.27 | 8.92 | 4.09 | 9.42 | 10.41 |
| | VidVRD-II | 3.57 | 8.59 | 12.39 | 5.70 | 13.22 | 18.34 | 7.35 | 18.84 | 26.44 |
| | RePro$^\dagger$ | 2.56 | 8.26 | 11.73 | 8.63 | 15.04 | 18.84 | 9.34 | 18.67 | 24.13 |
| | RePro | **6.10** | **13.38** | **16.52** | **10.32** | **19.17** | **25.28** | **12.74** | **25.12** | **33.88** |
| All | ALPro | 3.20 | 2.62 | 3.18 | 3.92 | 3.88 | 4.75 | 4.97 | 4.50 | 5.79 |
| | VidVRD-II | 12.74 | 9.90 | 12.59 | 17.26 | 14.93 | 19.68 | 19.73 | 18.17 | 24.90 |
| | RePro$^\dagger$ | 16.21 | 11.14 | 14.56 | 22.37 | 16.83 | 21.71 | 25.43 | 21.36 | 28.04 |
| | RePro | **21.33** | **12.92** | **15.94** | **30.15** | **19.75** | **25.00** | **34.90** | **25.50** | **32.49** |

## 4.4 EVALUATE OPEN-VOCABULARY RELATION CLASSIFICATION

The relation classification part of our RePro was trained separately by keeping the results of tracklet detection fixed. All of our experiments for relation classification used the same tracklet detection results (which is row **#4** in Table 1).

**Comparison to Conventional VidVRD SOTA Methods**. We compared our RePro with several SOTA methods in the conventional VidVRD setting, and showed the results in Table 2. The object tracklets and features used in SOTA methods are not uniform since VidVRD is a very challenging task (see Sec. A.6 for details). We can observe that even when our RePro is trained with only base category samples (while others are with both base and novel category samples), our performance on SGDet tasks is comparable to others'. When trained with both base and novel category samples, our RePro outperforms all other SOTA methods in all SGDet tasks and most RelTag tasks.

**Comparisons in the Setting of Open-VidVRD**. We compared the model performances in the setting of Open-VidVRD and showed results in Table 3. Since our RePro is the first Open-VidVRD method, we compared it to ALPro (implemented as Eq. (2)). We also re-implemented the SOTA method VidVRD-II (Shang et al., 2021) and trained it on base category samples. We replaced its classifier with text embeddings extracted by ALPro's text encoder. For both ALPro and VidVRD-II, we used a fixed (handcrafted) prompt "a video of relation [CLASS]". In addition, we reported the results of RePro's intuitive variant RePro$^\dagger$ as mentioned in the "**Discussion**" of Sec. 3.3.

From the results in Table 3, we can observe that our RePro outperforms ALPro, VidVRD-II and RePro$^\dagger$ by a large margin on both Novel-split and All-splits. By comparing RePro to ALPro, we show that, unlike that in tracklet classification, directly applying pre-trained VLM to relation classification is sub-optimal and achieves poor performance. By comparing RePro to VidVRD-II, we demonstrate the superiority of our prompt tuning framework over the fixed prompt design. By comparing RePro to RePro$^\dagger$, we validate the effectiveness of our training scheme for RePro.

## 4.5 ABLATION STUDIES

We conducted careful ablation studies as shown in Table 4. Since the compositional and motion-based prompt design is one of our main contributions, we conducted ablations w/o either of them (rows **#1, #2** and **#5**). To further show the effectiveness of our motion pattern design, we designed two variants, *i.e.*, rows **#3** (Ens) and **#4** (Rand). Their detailed settings are enumerated as follows: **#1**: It learns a single prompt representation $W_c$ as in Eq. (1). The obtained predicate text embedding has the half dimensions of $t_c^p$ in Eq. (6). So we calculated the visual embeddings of a tracklet pair as $v_{i,j} = v_i - v_j$ (different from the concatenated vector $v_{i,j}$ in Eq. (9)). **#2**: Training with

Table 4: Ablations (%) for RePro with different prompt design in VidVRD-test, where **C** stands for Compositional, and **M** stands for Motion cues. **Ens**: ensemble all the learned prompts by averaging their representations. **Rand**: readomly select a prompt without considering motion cues.

| | | C | M | SGDet | | | SGCls | | | PredCls | | |
|---|---|---|---|---|---|---|---|---|---|---|---|---|
| | | | | mAP | R@50 | R@100 | mAP | R@50 | R@100 | mAP | R@50 | R@100 |
| Novel-split | #1 | × | × | 3.50 | 9.91 | 13.88 | 7.21 | 14.54 | 19.83 | 8.63 | 20.33 | 27.43 |
| | #2 | ✓ | × | 5.57 | 11.40 | 14.87 | 10.31 | 16.52 | 21.81 | 11.83 | 22.31 | 30.90 |
| | #3 | ✓ | Ens | 6.24 | 11.57 | 15.20 | 10.77 | 16.03 | 21.98 | 12.36 | 21.32 | 29.91 |
| | #4 | ✓ | Rand | **7.14** | 11.90 | 14.87 | **10.85** | 16.52 | 23.30 | 12.42 | 22.64 | 30.90 |
| | #5 | ✓ | ✓ | 6.10 | **13.38** | **16.52** | 10.32 | **19.17** | **25.28** | **12.74** | **25.12** | **33.88** |
| All-splits | #1 | × | × | 19.73 | 12.26 | 15.36 | 26.80 | 18.24 | 23.06 | 30.80 | 23.70 | 30.42 |
| | #2 | ✓ | × | 18.47 | 11.95 | 15.28 | 25.52 | 18.13 | 23.12 | 29.45 | 23.39 | 30.17 |
| | #3 | ✓ | Ens | 20.15 | 12.38 | 15.61 | 27.93 | 18.61 | 23.55 | 31.68 | 23.61 | 30.29 |
| | #4 | ✓ | Rand | **21.72** | 12.71 | 15.78 | 29.15 | 19.15 | 24.13 | 33.11 | 24.38 | 31.49 |
| | #5 | ✓ | ✓ | 21.33 | **12.92** | **15.94** | **30.15** | **19.75** | **25.00** | **34.90** | **25.50** | **32.49** |

compositional prompt but without motion cues. **#3** & **#4**: Training with compositional prompt, but the prompt is randomly selected from the 6 groups without considering motion cues. For testing, the prompts are ensembled by averaging (**Ens**) or randomly selected (**Rand**). **#5**: The proposed RePro.

**Compositional Prompt**. By comparing the results in **#1** and **#2**, we can observe that the compositional prompt can effectively improve the performance on Novel-split. Meanwhile, the improvement on All-splits is not significant. We conjecture that the base relations, the majority of All-splits, require less compositional semantic contexts for prompt learning.

**Motion-based Prompt Groups**. By comparing our RePro (**#5**) vs. **#2**, we can observe that with the help of motion cues, our RePro achieves significant improvements of recall on all tasks, and also achieves considerable improvements in mAP on most tasks. We can see that the improvement on Novel-split is more

Table 5: Ablations (%) on VidOR-val.

| | C | M | SGCls | | PredCls | |
|---|---|---|---|---|---|---|
| | | | R@50 | R@100 | R@50 | R@100 |
| Novel-split | × | × | 0.86 | 0.86 | 2.30 | 2.88 |
| | ✓ | × | 1.72 | 2.59 | 6.62 | 8.06 |
| | ✓ | Ens | **2.30** | **2.59** | 7.20 | 8.35 |
| | ✓ | Rand | 2.01 | 2.30 | 5.76 | 7.20 |
| | ✓ | ✓ | 2.01 | 2.30 | **7.20** | **8.35** |
| All-splits | × | × | 9.49 | 12.85 | 25.62 | 34.83 |
| | ✓ | × | **10.06** | **13.40** | 27.00 | **36.73** |
| | ✓ | Ens | 9.49 | 12.68 | 25.66 | 35.16 |
| | ✓ | Rand | 10.03 | 13.13 | 26.94 | 36.48 |
| | ✓ | ✓ | 10.03 | 12.91 | **27.11** | 35.76 |

significant than that on All-splits, *e.g.*, 14.87%→16.52% vs. 15.28%→15.94% on R@100 of SGDet, showing that the motion-based prompt has a better generalizability for detecting novel relations. Besides, if comparing RePro (**#5**) to **Ens** (**#3**), we can see that RePro outperforms **Ens** in Novel-split on most tasks, and achieves considerable improvements for All-splits on all tasks. Compared to **Rand** (**#4**), RePro achieves clear improvements on most metrics for both Novel-split and All-splits.

**Ablations on VidOR**. We conducted the same ablation studies on VidOR-val, as shown in Table 5. Firstly, we can observe that the compositional prompt representation shows its efficiency on both Novel-split and All-splits, *e.g.*, 0.86%→1.72% and 9.49%→10.06% on R@50 of SGCls. For the motion-based prompt groups, the improvement of RePro is small due to the biased data distribution (Li et al., 2021), *i.e.*, the predicate categories strongly depend on the visual cues of subject and object tracklets, making the model predict relations simply based on object appearances without considering motion cues. More results on VidOR are left in the Appendix (Sec. A.7).

## 5 CONCLUSIONS

In this paper, we introduced the challenging Open-VidVRD task. We analyzed two key characteristics, *i.e.*, compositional and motion-related, when applying prompt tuning in this new task. We proposed a novel method called RePro that learns compositional prompt representations while considering motion-based contexts. Our evaluations on both conventional and open-vocabulary datasets show a clear superiority of RePro for tackling video visual relation detection tasks.

**Acknowledgement:** This work was supported by the National Key Research & Development Project of China (2021ZD0110700), the National Natural Science Foundation of China (U19B2043, 61976185), and the Fundamental Research Funds for the Central Universities (226-2022-00051). This work was also supported by A*STAR under its AME YIRG grant (Project No. A20E6c0101), and Singapore MOE Tier 2.

## ETHICS AND REPRODUCIBILITY STATEMENTS

**Ethics statement**. The open-vocabulary video visual relation detection (Open-VidVRD) that we introduced in this paper is a general extension of conventional VidVRD, and there are no known extra ethical issues in terms of the Open-VidVRD task and the proposed RePro model. As for the pre-trained visual-language model (VLM) used for Open-VidVRD, the large-scale pre-training data might contain some videos and captions involved with discrimination/bias issues. When applying pre-trained VLM to Open-VidVRD, the model tends to predict relations based more on the pre-trained knowledge, and focus less on the visual cues. For example, when the pre-trained data involved with unethical videos and captions, a model might predict "person-punch-dog" given a video of person caressing dog, which implies a person is abusing animals. To avoid the potential ethical issues, we can design algorithms to filter out those unethical training data for VLMs. For Open-VidVRD models, we can also introduce some common sense knowledge and design some rule-based methods to filter out those unreasonable relation triplets that involve ethical issues.

**Reproducibility Statement**. Our RePro is mainly implemented based on the realsed code of ALPro (Li et al., 2022a), VinVL (Zhang et al., 2021), and VidVRD-II (Shang et al., 2021). We first modified the code of VinVL to fit the video data and to extract object tracklets in each video. Then We modified the code of VidVRD-II to be compatible with the visual and text encoder of ALPro, and to fit the Open-VidVRD setting. We provide the the detailed base/novel split information of object and predicate categories in the Appendix (Sec. A.8) to ensure all experiments can be reproduced. When training the RePro model and its variants, we manually set the random seed and fixed the seed for all experiments to ensure they can be reproduced. We also provide the code of our RePro model and the training/evaluate scripts in the supplementary materials.

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

## A APPENDIX

This Appendix has the following contents:

- More details about the relative position feature are in Sec. A.1.
- More details about the motion pattern are in Sec. A.2.

- More details about the hyperparameters are given in Sec. A.3.
- Analysis of the performance improvement in different predicate groups are in Sec. A.4.
- Potential improvements of the motion pattern design are introduced in Sec. A.5.
- The detailed experiment settings of the compared SOTA methods are introduced in Sec. A.6.
- More experiment results on VidOR are provided at Sec. A.7.
- The detailed base/novel split information of object and predicate categories are in Sec. A.8.

## A.1 RELATIVE POSITION FEATURE FOR TRACKLET PAIRS

We compute the relative position between bounding boxes of subject-object tracklet pair $\langle T_i, T_j \rangle$ by following Shang et al. (2021). Specifically, we compute the relative position feature between subject-object for the beginning and ending bounding boxes of their temporal intersection. For the beginning bounding boxes of the subject and object, the position feature is calculated as:

$$\boldsymbol{f}_{i,j}^{B} = \left[ \frac{x_i - x_j}{x_j}, \frac{y_i - y_j}{y_j}, \log \frac{w_i}{w_j}, \log \frac{h_i}{h_j}, \log \frac{w_i h_i}{w_j h_j}, \frac{t_i - t_j}{L_{\text{seg}}}. \right], \tag{12}$$

where $(x_i, y_i)$ is the central coordinates of $T_i$'s beginning bounding box, $(w_i, h_i)$ is its width and height, and $t_i$ is the frame ID of this beginning bounding box. $(x_j, y_j, w_j, h_j, t_j)$ is defined similarly for $T_j$. $L_{\text{seg}}$ is the number of frames in each video segment, and following Shang et al. (2021), we set $L_{\text{seg}} = 30$. The relative position feature between the ending bounding boxes of $\langle T_i, T_j \rangle$ is defined similarly as $\boldsymbol{f}_{i,j}^{B}$, and is denoted as $\boldsymbol{f}_{i,j}^{E}$. The final relative position of $\langle T_i, T_j \rangle$ is concatenated as $\boldsymbol{f}_{i,j}^{s} = [\boldsymbol{f}_{i,j}^{B}, \boldsymbol{f}_{i,j}^{E}]$, and $\boldsymbol{f}_{i,j}^{s} \in \mathbb{R}^{12}$.

## A.2 DETAILS ABOUT THE MOTION PATTERNS

We provide a schematic of the motion patterns defined in Eq. (8), as shown in Figure 4. According to the definition in Eq. (8), $\boldsymbol{m}_{i,j}$ has only 6 possible values, i.e., $[+, -, +]$ and $[-, +, -]$ are impossible.

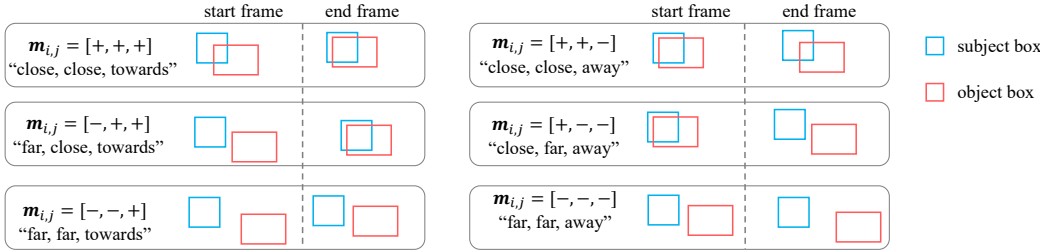

Figure 4: The schematic of the 6 motion patterns defined in Eq. (8).

## A.3 HYPERPARAMETERS

We set $\phi_o, \phi_p$ and $\phi_{\text{pos}}$ all as two-layer MLPs with hidden dimension 768. The $\lambda$ for weighting the distillation loss (i.e., Eq. (5)) was set as 5.0. The prompt length $L$ was set as 10. The softmax temperature $\tau$ was set as learnable. The GIoU threshold $\gamma$ was chosen based on the statistics of the training set, by making the tracklet pairs evenly distributed w.r.t different motion patterns. In our implementation, $\gamma$ was set as -0.3 for VidVRD and -0.25 for VidOR. We trained our RePro using Adam (Kingma & Ba, 2014) with a learning rate 1e-4, and stopped the training when SGDet mAP drops.

## A.4 ANALYSIS OF THE PERFORMANCE IMPROVEMENT IN DIFFERENT PREDICATE GROUPS

We evaluated the Recall@100 in the PredCls setting of some predicate groups (grouped by the prefix of predicate words) at the novel-split of the VidVRD dataset. We compared our RePro with the mean ensemble (Ens) and random select (Rand) variants of RePro (refer to Sec. 4.5). The results

show that the improvements of motion-related predicates are much larger than other context-related predicates. For example, we have 9.68% absolute improvements on "run" (e.g., "run past", "run next to"). While for those predicates that can be roughly inferred by the context (e.g., "fly", "swim"), our approach has limited contributions. This indicates that the performance improvements of our RePro are largely attributed to motion cues.

Table 6: Recall@100 (%) of PredCls on the test set of VidVRD w.r.t different predicate groups.

| Methods | move | sit | run | walk | stop | stand | fly | swim |
|---------|------|-----|-----|------|------|-------|-----|------|
| Ens | 34.48 | 50.92 | 12.90 | 18.30 | 37.03 | 35.51 | 37.50 | **15.38** |
| Rand | 37.93 | 51.85 | 16.12 | 18.30 | **44.44** | 36.44 | **50.00** | **15.38** |
| RePro | **44.82** | **55.55** | **25.80** | **18.95** | 40.47 | **41.12** | **50.00** | 12.82 |

## A.5 POTENTIAL IMPROVEMENTS OF THE MOTION PATTERN DESIGN

The current proposed GIoU-based motion pattern design can be further improved. Based on our proposed motion-pattern based prompt group learning framework, we can design other fancier motion capturing approaches, *e.g.*, automatically learning the motion primitives from the training set. Then for each test sample, the motion pattern can be decomposed as the weighted combination of motion primitives. Consequently we can use the weighted combination of the prompt representations as the desired prompt representation. We leave this as future work.

## A.6 DETAILED EXPERIMENTAL SETTINGS OF THE COMPARED SOTA METHODS

Since VidVRD is a very challenging task, The object tracklets and features used in SOTA methods in Table 2 are not uniform. The object tracking algorithms for tracklet generation include Seq-NMS (Han et al., 2016) and deepSORT (Wojke et al., 2017). The features include RoI Aligned features, I3D features (Carreira & Zisserman, 2017), and improved dense trajectory (iDT) features (Shang et al., 2017) Here we enumerate their details as follows:

- Su et al. (2020) uses Seq-NMS for tracklets generation, and improved dense trajectory (iDT) feature and relative motion feature of tracklet pairs for relation classification.
- Liu et al. (2020) uses deepSORT for tracklets generation, and uses RoI feature, I3D feature, and relative motion feature of tracklet pairs for relation classification.
- Li et al. (2021) uses Seq-NMS for tracklets generation, and uses RoI feature and relative motion feature of tracklet pairs for relation classification.
- Gao et al. (2022) uses deepSORT for tracklets generation, and uses RoI feature, I3D feature for relation classification.
- Our RePro uses Seq-NMS for tracklets generation, and uses RoI feature and relative motion feature of tracklet pairs for relation classification.

## A.7 MORE EXPERIMENT RESULTS ON VIDOR

Table 7: Performance (%) on the validation set of VidOR.

| Methods | Novel-split | | | | All-splits | | | |
|---------|-------------|------|------|------|------------|------|------|------|
| | SGCls | | PredCls | | SGCls | | PredCls | |
| | R@50 | R@100 | R@50 | R@100 | R@50 | R@100 | R@50 | R@100 |
| ALPro | 3.17 | 3.74 | 8.35 | 9.79 | 0.95 | 1.32 | 2.61 | 3.66 |
| VidVRD-II | 1.44 | 2.01 | 4.32 | 4.89 | 9.40 | 12.78 | 24.81 | 34.11 |
| RePro[†] | 1.72 | 2.30 | 6.62 | 8.06 | 8.88 | 11.52 | 23.84 | 31.57 |
| RePro | **2.01** | **2.30** | **7.20** | **8.35** | **10.03** | **12.91** | **27.11** | **35.76** |

We provide more experiment results for our RePro on the validation set of VidOR, as shown in Table 7. We first compare our RePro with using ALPro directly perform relation classification as Eq. (2). We find that ALPro performs slightly better than RePro on novel-split, because ALPro

doesn't have the trend of fitting base categories. However, ALPro performs much worse than Re-Pro in All-splits due to not trained on base categories. Furthermore, we also compare RePro with the VidVRD-II (Shang et al., 2021) baseline and the variant RePro†. We can observe that RePro outperfoms both VidVRD-II and RePro† by a large margin on both novel-split and all-split.

## A.8 Detailed base/novel categories of object and predicate for VidVRD and VidOR

We list the base/novel categories of object and predicate for training and evaluating our RePro and other baselines in all experiments. We also provide more statistics information in the supplementary materials

**VidVRD Object**

25 base object categories:

> "airplane", "bicycle", "bird", "bus", "car", "dog", "domestic_cat", "elephant", "hamster", "lion", "monkey", "rabbit", "sheep", "snake", "squirrel", "tiger", "train", "turtle", "whale", "zebra", "ball", "frisbee", "sofa", "skateboard", "person"

10 novel object categories

> "horse", "watercraft", "giant_panda", "fox", "red_panda", "cattle", "motorcycle", "bear", "antelope", "lizard"

**VidVRD Predicate**

71 base predicate categories:

> "behind", "chase", "creep_behind", "creep_beneath", "creep_front", "creep_left", "creep_right", "fall_off", "faster", "fly_above", "fly_next_to", "fly_past", "fly_toward", "fly_with", "follow", "front", "jump_beneath", "jump_front", "jump_left", "jump_next_to", "jump_right", "jump_toward", "larger", "left", "lie_behind", "lie_front", "lie_left", "lie_next_to", "lie_right", "move_behind", "move_beneath", "move_front", "move_left", "move_right", "move_with", "next_to", "play", "ride", "right", "run_behind", "run_front", "run_left", "run_past", "run_right", "run_with", "sit_above", "sit_front", "sit_left", "sit_right", "stand_behind", "stand_front", "stand_left", "stand_next_to", "stand_right", "stop_behind", "stop_front", "stop_left", "stop_right", "swim_front", "swim_left", "swim_right", "swim_with", "taller", "touch", "walk_behind", "walk_front", "walk_left", "walk_next_to", "walk_right", "walk_with", "watch"

61 novel predicate categories:

> "above", "away", "beneath", "bite", "creep_above", "creep_away", "creep_next_to", "creep_past", "creep_toward", "drive", "feed", "fight", "fly_away", "fly_behind", "fly_front", "fly_left", "fly_right", "hold", "jump_above", "jump_away", "jump_behind", "jump_past", "jump_with", "kick", "lie_above", "lie_beneath", "lie_inside", "lie_with", "move_above", "move_away", "move_next_to", "move_past", "move_toward", "past", "pull", "run_above", "run_away", "run_beneath", "run_next_to", "run_toward", "sit_behind", "sit_beneath", "sit_inside", "sit_next_to", "stand_above", "stand_beneath", "stand_inside", "stand_with", "stop_above", "stop_beneath", "stop_next_to", "stop_with", "swim_behind", "swim_beneath", "swim_next_to", "toward", "walk_above", "walk_away", "walk_beneath", "walk_past", "walk_toward"

**VidOR Object**

50 base object categories:

> "adult", "child", "toy", "dog", "baby", "car", "chair", "table", "sofa", "ball/sports_ball", "screen/monitor", "cup", "bicycle", "guitar", "bottle", "backpack", "handbag", "baby_seat", "camera", "cat", "cellphone", "bird", "sheep/goat", "laptop", "ski", "stool", "watercraft", "duck", "bus/truck", "bench", "fruits", "baby_walker", "horse", "bat", "dish",

"electric_fan", "kangaroo", "motorcycle", "lion", "hamster/rat", "refrigerator", "elephant", "faucet", "cake", "penguin", "sink", "piano", "microwave", "cattle/cow", "aircraft"

30 novel object categories:

"antelope", "vegetables", "panda", "rabbit", "fish", "train", "snowboard", "suitcase", "squirrel", "leopard", "chicken", "skateboard", "traffic_light", "surfboard", "camel", "racket", "bread", "bear", "oven", "scooter", "frisbee", "stop_sign", "turtle", "stingray", "pig", "crab", "crocodile", "toilet", "tiger", "snake"

**VidOR Predicate**

30 base predicate categories:

"next_to", "in_front_of", "watch", "behind", "away", "towards", "beneath", "above", "hold", "lean_on", "speak_to", "ride", "hug", "touch", "carry", "hold_hand_of", "bite", "push", "pull", "play(instrument)", "grab", "release", "pat", "inside", "lift", "caress", "point_to", "press", "hit", "use"

20 novel predicate categories:

"kick", "chase", "wave", "smell", "throw", "feed", "kiss", "wave_hand_to", "shout_at", "drive", "clean", "lick", "squeeze", "shake_hand_with", "get_off", "knock", "cut", "open", "get_on", "close"

