# OpenReview forum: "Compositional Prompt Tuning with Motion Cues for Open-vocabulary Video Relation Detection"
_ICLR.cc/2023/Conference — ICLR 2023 poster_

### Official Review · Reviewer_K5Cm · 2022-10-24

**Confidence:** 4
**Clarity, Quality, Novelty And Reproducibility:** The proposed learnable compositional …
**Correctness:** 4
**Technical Novelty And Significance:** 3
**Empirical Novelty And Significance:** 2
**Recommendation:** 6

**Strength And Weaknesses:**

Strengths:
1. The authors propose to add the composition capability and motion cues to the prompt, making it more robust and less biased for video relation detection. The idea is new.
2. The design of multi-mode (multiple different motion patterns) prompt groups is interesting.
3. The paper is well-written and easy to follow.
4. The authors conducted several experiments to verify the effectiveness of their method. It shows comparable results when comparing to VidVRD SOTA. In their ablation study, their open-vocabulary-based results are better than the baselines.

Weaknesses:
1. The motion pattern design seems to be complex and not easy to handle a variety of motions. While the use of GIoU is nice, the current design considered only toward/away and near/far. It is difficult to extend to generic motions with GIoU. Since the current design of motion group is very limited, the strong performance improvements are less convincing to be attributed to motion cues.
2. In the comparison with traditional VidVRD, one may wonder the performance of the proposed model if adding novel samples.


**Summary Of The Paper:**

This paper presents an interesting prompt tuning method for video relation detection. It includes compositional and motion related cues.

The authors also propose a new task, called Open VIdVRD, which further considers unseen relationship combinations and unseen objects/motions.

**Summary Of The Review:**

Overall the proposed method is novel and interesting. However, the design of motion cues seems limited in current stage. It still requires hand-craft effort or expert knowledge.

The introduced task is a good addition to VidVRD dataset. Results on open-vocabulary setting are good compared to baseline variants.

---

> ### Author Response · Authors · 2022-11-16
> **Response to Reviewer K5Cm**
>
> Thank you for the detailed comments. We are willing to address your questions.
>
> $\quad$
>
> ## Q1: The improvements of motion group are less convincing.
> > Since the current design of motion group is very limited, the strong performance improvements are less convincing to be attributed to motion cues.
>
> **A1:** As for the current GIoU-based motion patterns, the performance improvement attributed to motion cues is reasonable. Here we evaluated the Recall@100 of some predicate groups (grouped by the prefix of predicate words) at the novel-split.
>
>
> | | move | sit | run | walk | stop | stand | fly | swim |
> | ----- | --------- | --------- | --------- | --------- | --------- | --------- | --------- | --------- |
> | Ens | 34.48 | 50.92 | 12.90 | 18.30 | 37.03 | 35.51 | 37.50 | 15.38 |
> | Rand | 37.93 | 51.85 | 16.12 | 18.30 | **44.44** | 36.44 | 50.00 | **15.38** |
> | RePro | **44.82** | **55.55** | **25.80** | **18.95** | 40.47 | **41.12** | **50.00** | 12.82 |
>
> We compared our RePro with the mean ensemble (Ens) and random selected (Rand) variants of RePro (refer to Sec. 4.5). In Ens, the prompt is ensembled by averaging representations. In Rand, the prompt is randomly selected without considering motion patterns.
>
> The results show that the improvements of motion-related predicates is much larger than other context-related predicates. For example, we have 9.68 absolute percentage points improvement on "run" (e.g., "run past, run next to"). While for those predicates that can be roughly inferred by the context (e.g., fly, swim), our approach has little contribution. Therefore, the performance improvements are largely attributed to motion cues.
>
> We have added the above results in the Appendix of the revised paper.
>
> **Corresponding Revision:** Sec A.4 (**Lines 510 ~ 518 and Table 6**) of the revised Appendix
>
> $\quad$
>
> ## Q2: The design of motion cues is limited.
> > The motion pattern design seems to be complex and not easy to handle a variety of motions. While the use of GIoU is nice, the current design considered only toward/away and near/far. It is difficult to extend to generic motions with GIoU.
> >
> > However, the design of motion cues seems limited in current stage. It still requires hand-craft effort or expert knowledge.
>
> **A2:**  Indeed, the current proposed GIoU-based design is not perfect. It's just a simple and intuitive way to calculate motion patterns. Our main emphasis lies in the motion-pattern-based prompt group learning framework. Under this framework, we can design other fancier motion capturing approaches. For example, 1) we first automatically learn motion primitives from the training set. 2) Then, for each test sample, its motion pattern can be decomposed as the weighted combination of motion primitives. 3) Consequently, we can use the weighted combination of the prompt representations as the desired prompt representation.
>
> We leave the above approach as future work.  We have added this in the Appendix of the revised paper.
>
> **Corresponding Revision:** Sec. A.5 (**Lines 519 ~ 525**) in the revised Appendix
>
> $\quad$
>
> ## Q3: More experimental results of the proposed method by adding novel samples
> > In the comparison with traditional VidVRD, one may wonder the performance of the proposed model if adding novel samples.
>
> **A3:** Thank you for your suggestion. We have conducted additional experiments to show the performance of our RePro when trained on both base and novel samples:
>
> |      Methods      | Training Data |    mAP    |    R@50   |   R@100   |    P@1    |    P@5    |    P@10   |
> |:-----------------:|:-------------:|:---------:|:---------:|:---------:|:---------:|:---------:|:---------:|
> |  Su et al. (2020) |   base+novel  |   19.03   |    9.53   |   10.38   |   57.50   |   41.40   |   29.45   |
> | Liu et al. (2020) |   base+novel  |   18.38   |   11.21   |   13.69   |   60.00   |   43.10   |   32.24   |
> |  Li et al. (2021) |   base+novel  | **22.97** |   12.40   |   14.46   | **68.83** | **49.87** | **35.57** |
> |  Gao et al (2022) |   base+novel  |   17.67   |    9.63   |   11.29   |   56.00   |   43.80   |   32.85   |
> |    RePro (Ours)   |      base     |   21.72   | **12.61** | **15.84** |   61.50   |   43.70   |   31.92   |
> |    RePro (Ours)   |   base+novel  | **25.02** | **13.81** | **17.22** | **67.50** | **50.10** | **36.50** |
>
> From the above results, we can observe that our RePro outperforms all other SOTA methods in all metrics when trained with both base and novel category samples.
>
> We have added the above results in Table 2 of the revised paper.
>
> **Corresponding Revision:** Sec. 4.4 ( last row of **Table 2**) in the revised paper

---

> > ### Comment · Reviewer_K5Cm · 2022-11-28
> > **Reponse**
> >
> > Thank you for the additional experiments, which better addressed my previous concerns. I will keep my rating as weak accept. The additional discussions and results should be added to the final paper or supplementary material.

---

> > > ### Author Response · Authors · 2022-11-29
> > > **Reponse to Reviewer K5Cm**
> > >
> > > Thank you for your reply. We are glad to solve your concerns. The above results have been added in the revised paper (last row of Table 2) and the revised Appendix ( Sec A.4, Lines 510 ~ 518 and Table 6). Please kindly take a look. We will continue to refine the discussion in the final version of the paper and appendix.
> > >
> > > We are willing to have further discussion with you. Please feel free to leave comments if you have any other questions.

---

### Official Review · Reviewer_Nj5q · 2022-10-24

**Confidence:** 3
**Correctness:** 3
**Technical Novelty And Significance:** 3
**Empirical Novelty And Significance:** 3
**Recommendation:** 6

**Clarity, Quality, Novelty And Reproducibility:**

- The main idea is interesting and the work is well-written and easy to follow.
- The proposed method is clearly described and the results has potential impact.

- They partially shared their code, and hyper parameter settings. The results are reproducible.


**Strength And Weaknesses:**

Strength
- They proposes a novel setting for video visual relation detection task, Open-VidVRD.
- They clearly describe interesting idea in this paper. The compositional prompt representation learning method that models the prompt contexts for subject and object separately. the motion-cue-based prompt groups shows good generalization ability on the tasks.

Weakness
- The novelty is limited. The paper claims that the first one to research on the the open vocabulary video visual relation detection, but the open vocabulary VRD has been widely studied in visual domain, which potentially could be compared.
- The motivations in evaluation section, and ablations of the proposed approach is not easy to follow.
It's also hard to follow how the proposed approach improves the performance on the Open-VidVRD benchmark.

**Summary Of The Paper:**

This paper proposes a new approach to video visual relation detection that is based on extension of compositional prompt tuning with motion cues. The Relation Prompt (RePro) is designed to address the technical challenges of Open-vocabulary Video Visual Relation Detection: the prompt tokens should respect the two different semantic roles of subject and object while the tuning should account for the diverse spatiotemporal motion patterns of the subject-object compositions. They achieves a new state-of-the-art performance on two VidVRD benchmarks of not only the base training object and predicate categories, but also the unseen ones.

**Summary Of The Review:**

This paper proposes a new method for several technical challenges of Open-vocabulary Video Visual Relation Detection, and also provides useful prompt group that can be generalized. The paper shows the video visual relation detection is generalizable with the proposed framework. The proposed video visual relation detection task is very interesting, but I would appreciate if the author can clarify the evaluation and motivation behind the design. Overall, I lean toward acceptance.

---

> ### Author Response · Authors · 2022-11-16
> **Response to Reviewer Nj5q (1/2)**
>
> Thank you for the detailed comments. We are willing to address your questions.
>
>
> ## Q1: Limited novelty.
> > The novelty is limited. The paper claims that the first one to research on the the open vocabulary video visual relation detection, but the open vocabulary VRD has been widely studied in visual domain, which potentially could be compared.
>
> **A1:** To the best of our knowledge, **we are the first one to explore the open vocabulary setting of VIDEO VRD**, although there are some existing works based on related settings (as mentioned in Sec. 2). Here we would like to do more discussions about the differences between theirs and ours:
>
>
> - In video VRD, existing works like [1,2] study the zero-shot setting only on unseen object-predicate combinations, in which the objects and predicates are seen in the training set. In contrast, our method deals with the object and predicate categories that are totally unseen in the dataset, which had been discussed in Section 2.
>
> - In image VRD, existing works [3,4] only consider unseen combinations, in which the objects and predicates are seen in the training set. There is another concurrent work [5] (which released on arXiv on August 8, 2022) proposes an open-vocabulary setting in image VRD. We had discussed this work and compared it with ours in Section 2. In this work, the authors put their emphasis on unseen object categories, and learn the prompt representations from the visual cues of static subject and object.
>
> In addition, it is worth noting that our Open-VidVRD is w.r.t the video level, where the visual relations are between object tracklets, rather than between each object bounding box in frame-level. It is thus nontrivial to directly apply zero-shot image VRD methods to our Open-VidVRD setting.
>
> In the revised paper, we have added an extra paragraph in the Related Works Section to discuss the above differences.
>
> **Corresponding Revision:** Sec. 2 (**Lines 102 ~ 111**) in the revised paper.
>
> [1] Xindi Shang, et al. Video visual relation detection. In *ACM MM*, pp. 1300–1308, 2017.
>
> [2] Yicong Li, et al. Interventional video relation detection. In *ACM MM*, pp. 4091–4099, 2021.
>
> [3] Tao He, et al. Towards open-vocabulary scene graph generation with prompt-based finetuning. In *ECCV*, 2022.
>
> [4] Kaihua Tang, et al. Unbiased scene graph generation from biased training. In *CVPR*, pp. 3716–3725, 2020.
>
> [5] Xuan Kan, et al. Zero-shot scene graph relation prediction through common-sense knowledge integration. In *Joint European Conference on Machine Learning and Knowledge Discovery in Databases*, pp. 466–482, 2021

---

> > ### Author Response · Authors · 2022-11-16
> > **Response to Reviewer Nj5q (2/2)**
> >
> >
> > ## Q2: Clarification for the unclear evaluation and motivation.
> > > The motivations in evaluation section, and ablations of the proposed approach is not easy to follow. It's also hard to follow how the proposed approach improves the performance on the Open-VidVRD benchmark.
> > > I would appreciate if the author can clarify the evaluation and motivation behind the design
> >
> > **A2:**   Thank you for the suggestions. We clarify our motivations behind the evaluation and ablations as follows. Please note that we have revised our manuscript accordingly.
> >
> >
> > **The evaluation of open-vocabulary tracklet classification (Table 1)**:
> > - **Motivation for comparing RePro vs. ALPro**: A straightforward baseline is directly applying the pre-trained VLM (e.g., ALPro in our case) to perform classification as in Eq. (2). This baseline has a significant computational overhead due to the heavy pipeline of ALPro's visual encoder.  We thus compared our RePro with ALPro to show that RePro can achieve comparable performances with much less computational costs by using the projection layer $\phi_o$.
> >
> >
> > - **Motivation for the ablation of distillation**: The projection layer $\phi_o$ is learned by visual distillation (i.e., Eq. (5)). We thus conducted the ablation study for this visual distillation (rows #2 vs. #3 in Table 1).
> >
> >
> > - **Motivation for verifying $\mathcal{L}_{cls-neg}$** : How to model the background sample is a key challenge and has been discussed a lot in many open-vocabulary object detection works. There are two typical approaches: 1) using a unique background embedding in addition to the class text embeddings (i.e., using a ($C+1$)-d classifier), and 2) using only the $C$ class text embeddings, i.e., a $C$-d classifier, and computing the loss of background sample as $\mathcal{L}_\{cls-neg}$ in Eq. (4), which forces the prediction (from any negative sample) on each base class to be $1/|\mathcal{C}_b|$ . Therefore, we design the ablation study to verify the effectiveness of $\mathcal{L}_\{cls-neg}$ (rows #3 vs. #4 in Table 1).
> >
> > **The evaluation of open-vocabulary relation classification (Table 3):**
> >
> >
> > - **Motivation for comparing RePro vs. ALPro**: We want to show that, unlike that in tracklet classification, directly applying pre-trained ALPro in relation classification is sub-optimal and achieves poor performance. This further shows the importance of our prompt design tailored for dynamic visual relations.
> >
> >
> > - **Motivation for comparing RePro vs. VidVRD-II**: We re-implemented the SOTA method VidVRD-II and replaced its classifier with text embeddings extracted by the ALPro’s text encoder, with a fixed prompt "a video of relation [CLASS]". By comparing RePro with VidVRD-II , we demonstrate the superiority of our prompt tuning framework (in RePro) over the fixed (handcraft) prompt design.
> >
> >
> > - **Motivation for comparing RePro vs. RePro$^\dagger$**: as mentioned in the "Discussion" of Section 3.3, this is to show the effectiveness of our training scheme designed for RePro.
> >
> >
> >
> >
> > **For the ablations in Section 4.5 (Table 5):**
> >
> >
> > - **Motivation for the ablation of w/o C, w/o M**: The compositional- and motion-based prompt design is one of our main contributions. We conduct ablations w/o either of them. 1) Results of row-#1&2 show the effectiveness of the *compositional* design. 2) Results of row-#2&5 show the effectiveness of the *motion* design.
> >
> >
> > - **Motivation for the ablation of Ens and Rand**: To further show the effectiveness of our motion pattern design, we show the results of two naive variants respectively on row-#3 (Ens) and row-#4 (Rand). In Ens, the prompt is ensembled by averaging representations. In Rand, the prompt is randomly selected without considering motion patterns. The results of row-#5 compared with row-#3&4 show the effectiveness of our GIoU based motion pattern selection.
> >
> > We have revised our writing to highlight the above motivations more clearly in the new version of the paper.
> >
> > **Corresponding Revision:**  Sec. 4.3 (**Lines 272 ~ 283, Lines 292 ~ 295**), Sec. 4.4 (**Lines 297 ~ 319**) , and  Sec. 4.5 (Lines **321 ~ 324**) in the revised paper.

---

> > > ### Comment · Reviewer_Nj5q · 2022-12-04
> > > **Thanks**
> > >
> > > I appreciate the authors' efforts to respond to my points.
> > > The authors addressed all of my questions, and I will further discuss with other reviewer. I will keep my previous rating. 6: marginally above the acceptance threshold

---

> > > > ### Author Response · Authors · 2022-12-05
> > > > **Response to Reviewer Nj5q**
> > > >
> > > > Thank you for acknowledging our response. Please feel free to leave comments if you have any other concerns or questions.

---

### Official Review · Reviewer_Fty6 · 2022-10-25

**Confidence:** 3
**Correctness:** 3
**Technical Novelty And Significance:** 3
**Empirical Novelty And Significance:** 3
**Recommendation:** 6

**Clarity, Quality, Novelty And Reproducibility:**

**Clarity, Quality, Novelty:**
The paper is well-written, and it shows enough novelty.

**Reproducibility:**
The authors added a separate section to discuss reproducibility. They also provide the code, but this reviewer hasn't checked the code.


**Strength And Weaknesses:**

**Strengths:**

* The paper is well written.

* The authors have introduced an interesting problem and proposed a novel solution for the problem. Further, they showed rich experimental results.

**Weaknesses:**

* What is the difference between visual embeddings ($\mathbf{v}_i^O$) and RoI Aligned features ($\mathbf{f}_i$) in Sections 3.1 and 3.2?

* Figure 3 mix-uses texts (e.g. "dog", "child") for both visual features and categories.

* The experimental settings in Section 4.4 is not clear. Did the authors use the same tracklets for the comparisons in Tables 3 and 4? This reviewer is unsure if the comparisons are valid.

* Typos:
    * "we distil the knowledge ..." in Line 172



**Summary Of The Paper:**

The paper introduces a new task called Open-vocabulary VidVRD, where the aim is to detect not only unseen relationships but also unseen objects and predicates. They propose a novel pipeline (i.e. RePro) to detect tracklets and classify the relations between tracklets. In particular, the proposed approach is a compositional and motion-based relation prompt learning framework. In the experiments, the authors show rich experiments, including various ablation studies.

**Summary Of The Review:**

The paper introduces an interesting task and a framework to solve the task.
Some parts are still unclear, and this reviewer left the questions in the Weaknesses section.

---

> ### Author Response · Authors · 2022-11-16
> **Response to Reviewer Fty6 (1/2)**
>
> Thank you for the detailed comments. We are willing to address all your questions.
>
> ## Q1: Unclear presentation about visual features.
>
> > What is the difference between visual embeddings ($\mathbf{v}_i^o$) and RoI Aligned features ($\mathbf{f}_i$) in Sections 3.1 and 3.2?
>
>  **A1:** These two features are different kinds of visual features: **RoI Aligned feature $\mathbf{f}_i$:** $\mathbf{f}_i$ is obtained from the object detector according to the tracklet regions. It is widely used as object's appearance feature in relation detect works (i.e., it serves as the *input data* of the Open-VidVRD task). **Visual embedding $\mathbf{v}_i$:** $\mathbf{v}_i$ is extracted by the visual encoder of VLM. It contains the knowledge from the pre-trained VLM, and serves as the *intermediate data* for transferring the open-vocabulary knowledge. We modified the caption of Figure 3 for a clearer explanation.
>
> **Illustration of Figure 3.** As shown in Figure 3, the visual embedding is only used at *training* time, in which we train two V2L project modules (i.e., $\phi_o$ and $\phi_p$) to transfer the knowledge from VLM to our RePro model. At test time, only $\mathbf{f}_i$ is used. For tracklet detection, the knowledge is transferred by aligning $\phi_o(\mathbf{f}_i)$ to $\mathbf{v}_i$, as in Eq.(5)
>
> For relation classification, the knowledge is transferred by the prompt representations learned in the supervision of $\mathbf{v}_{i,j}$ (i.e., the tracklet pair's visual embedding).
>
> **Implementation.**  In our implementation, we used ALPro as VLM, and accordingly $\mathbf{v}_i^o$ is a 256-d vector for each tracklet. For the RoI Aligned feature, we used Faster-RCNN, and accordingly $\mathbf{f}_i$ is a 2048-d vector for each tracklet.
>
> Note that we have simplified the notation $\mathbf{v}^o_i$ to $\mathbf{v}\_i$ (and $\mathbf{v}^p_{i,j}$ to $\mathbf{v}_{i,j}$) in the revised manuscript for more concise presentation.
>
> **Corresponding Revision:**  Sec. 3 ( **Lines 136 ~ 137, the caption of Figure. 3**) in the revised paper.
>
> $\quad$
>
> ## Q2: Unclear illustration of Figure 3.
> > Figure 3 mix-uses texts (e.g. "dog", "child") for both visual features and categories.
>
> **A2:**  Thank you for pointing out this problem. The "dog" for text embedding represents the object category of "dog", while the "dog" for the visual part represents the dog's visual embedding and RoI feature.
>
> **Corresponding Revision:**  In the revised paper, we have updated **Figure 3**, in which the text "dog" of visual feature has been removed and replaced with the notation $\mathbf{v}_i$.

---

> > ### Author Response · Authors · 2022-11-16
> > **Response to Reviewer Fty6 (2/2)**
> >
> >
> > ## Q3:  Unclear presentation of experimental settings.
> > > The experimental settings in Section 4.4 is not clear.
> >
> > **A3:** Thank you for your detailed comments. We have modified the writing of the experimental settings (both Table 2 and Table 3) in Section 4.4 in the revised paper (cf. **Lines 297 ~ 319**). Based on the revised version, we briefly explain them as follows:
> >
> > **Experimental Settings of Table 2**. Apart from the training data, other settings of object tracklets are not uniform since VidVRD is a very challenging task.
> >
> > - Su et al. (2020) uses Seq-NMS for tracklets generation, and improved dense trajectory (iDT) feature and relative motion feature of tracklet pairs for relation classification.
> >
> > - Liu et al. (2020) uses deepSORT for tracklets generation, and uses RoI feature, I3D feature, and relative motion feature of tracklet pairs for relation classification.
> >
> > - Li et al. (2021) uses Seq-NMS for tracklets generation, and uses RoI feature and relative motion feature of tracklet pairs for relation classification.
> >
> > - Gao et al. (2022) uses deepSORT for tracklets generation, and uses RoI feature, I3D feature for relation classification.
> >
> > - Our RePro uses Seq-NMS for tracklets generation, and uses RoI feature and relative motion feature of tracklet pairs for relation classification.
> >
> >
> > Note that Seq-NMS is a more naive method than deepSORT. We have added the above details in the revised Appendix (Sec A.6)
> >
> > **Conclusions from Table2.** In conclusion, the comparisons in Table 2 are not fair to some extent. Nevertheless, our RePro achieves comparable performance with only base category data training. When trained with both base and novel data, our RePro outperforms all other methods as shown in Table 2 of the revised paper.
> >
> > **Experimental Settings of Table 3**:
> >
> >
> > - For  ALPro,  we directly used the visual embedding and text embedding of ALPro to perform predicate classification as Eq. (2).
> > - For VidVRD-II, we replaced its classifier with text embeddings extracted by the ALPro’s text encoder. We retrained the VidVRD-II model while keep its classifier fixed.
> > - For both ALPro and VidVRD-II, we used fixed (handcraft) prompt like "a video of relation [CLASS]".
> >
> > **Conclusions from Table3.** By comparing ALPro with our RePro, we show that directly applying pre-trained VLM to Open-VidVRD is sub-optimal and achieves poor performance. By comparing VidVRD-II with RePro, we demonstrate the superiority of our prompt tuning framework over the VidVRD-II baseline.
> >
> > **Corresponding Revision:**  Sec 4.4 (**Lines 297 ~ 319**) of the revised paper, and Sec. A.6 (**Lines 526 ~ 541**) of the revised Appendix.
> >
> >  $\quad$
> >
> >
> > ## Q4: Clarification for the valid comparisons.
> > > Did the authors use the same tracklets for the comparisons in Tables 3 and 4? This reviewer is unsure if the comparisons are valid.
> >
> > **A4:**  Yes, they are the same. We believe that the comparisons in Table 3 and Table 4 are valid. Reasons are as follows: 1) The relation classification part of our RePro is trained separately by keeping the tracklet detection results fixed. 2) All of the experiments in Table 3 and Table 4  use the same tracklet detection results (both for the position and classification), which are on the row #4 of Table 1.
> >
> > To make these clearer, in the revised paper, we have highlighted the tracklet settings.
> >
> > **Corresponding Revision:**  Sec. 4.4 (**Lines 297 ~ 299**) in the revised paper.
> >
> >
> >  $\quad$
> >
> > ## Q5: Typos
> > > Typos:"we distil the knowledge ..." in Line 172
> >
> > **A5:** Thank you for pointing out these problems in detail. We have checked that the spelling of "distil" is a variant of "distill". For consistency, we have modified it as "distill" in the revised paper. We have also fixed other typos in the revised paper.
> >
> > **Corresponding Revision:**
> >
> > - In **Line173**: "we distil the knowledge ..." --> "we distill the knowledge ..."
> > - In **Eq.(5)** and **Line175**: $\mathcal{L}\_{distil}$ --> $\mathcal{L}_{distill}$
> > - In **Line187**: "where $\mathbf{s}_i$ and $\mathbf{o}_i$ is the learnable context vector" --> "where $\mathbf{s}_i$ and $\mathbf{o}_i$ are the learnable context vectors"
> > - In **Line201**: "... a simple and intuitive way to calculating" --> "... a simple and intuitive way to calculate"

---

> > > ### Comment · Reviewer_Fty6 · 2022-12-01
> > > **Response**
> > >
> > > Thanks for your response and modifying the manuscript with more details. I will keep my original rating (6: marginally above the acceptance threshold).

---

> > > > ### Author Response · Authors · 2022-12-02
> > > > **Response to Reviewer Fty6**
> > > >
> > > > Thank you for acknowledging our response and the modified manuscript. We would appreciate that if you could provide more details of unresolved concerns to help us further improve the paper. Thank you !

---

### Official Review · Reviewer_QLPW · 2022-10-25

**Confidence:** 5
**Correctness:** 4
**Technical Novelty And Significance:** 3
**Empirical Novelty And Significance:** 3
**Recommendation:** 6

**Clarity, Quality, Novelty And Reproducibility:**

Clarity : OK

Quality: good

Novelty: good

Reproducibility: the writing is not ideal, which makes it a bit difficult to reproduce, unless the author will release the code.

**Strength And Weaknesses:**

Strength

- The idea is simple, using compositional prompt and exploiting motion information do generate prompt do make sense in the video visual relation detection task.

- The results have clearly validate the claims, demonstrating state-of-the-art performance under various settings.

Weakness:

- The writing is not satisfactory, it can surely be better.

- Missing many references from ECCV:
[1] https://arxiv.org/abs/2112.04478
[2] https://arxiv.org/pdf/2208.03550
[3] https://arxiv.org/abs/2208.02816


**Summary Of The Paper:**

- The paper considers the problem of video visual relation detection under an open-vocabulary scenario.

- The main contribution is on prompt learning to adapt the strong visual-language model towards downstream tasks.

- To enable the open-vocabulary, the prompt learning has exploited two techniques, namely, the compositional prompt representation, and motion-based prompt groups, which makes sense.

- The idea has been evaluated on various datasets, and show clear performance improvements.

**Summary Of The Review:**

Overall, I think I like this paper, I like the motivation for solving downstream video tasks with prompt learning, the idea of using composition prompt generation for predicate is good, and exploiting motions makes sense in video understanding tasks.

However, the paper writing needs a lot improvement, now it's very hard to grasp the key ideas, too many notions, and unnecessary equations.

---

> ### Author Response · Authors · 2022-11-16
> **Response to  Reviewer QLPW (1/2)**
>
> Thank you for the detailed comments. We are willing to address all the mentioned weaknesses.
>
>
>
> ## Q1: The paper writing needs to be further improved.
> > The writing is not satisfactory, it can surely be better.
> > ... the paper writing needs a lot improvement
>
> **A1:** Thank you for the comments. For paper writing, we have made several improvements in the revised manuscript. The details are as follows:
>
> - In the Introduction section,  we improved the clarification of our key ideas (**Lines 76 ~ 78, Lines 80 ~ 82, Line 85**).
> - In the Related Work section,  we improved the writing to clarify the novelty of our Open-VidVRD setting compared to existing works with zero-shot setting (**Lines 102 ~ 111**).  We also included some references from ECCV which are related to our work (**Lines 112 ~ 115, Lines 118 ~ 120**).
> - In the Method section, we updated **Figure 3** and its caption for clearer presentation. We also simplified the notations and equations.
> - In the Experiments section, we improved the clarification of our motivations behind the evaluation and ablations (**Lines 272 ~  283, Lines 292 ~ 299, Lines 302 ~ 306**, and **Lines 308 ~ 324**). We also updated the new results of our RePro in conventional VidVRD setting trained on both base and novel category samples (**Table 2**).
>
> $\quad$
>
> ## Q2: Some references from ECCV are missing.
> > Missing many references from ECCV: [1] https://arxiv.org/abs/2112.04478 [2] https://arxiv.org/pdf/2208.03550 [3] https://arxiv.org/abs/2208.02816
>
> **A2:**  Thank you for the suggestions. It's worth noting that these three mentioned works mainly focus on video action recognition that is to classify a video into a single action class. In contrast,  VidVRD requires classifying the visual relations (which are combinational classes) in each object tracklet pair (which can be multiple in a video).
>
> In the revised paper, we have added more detailed discussion and comparisons (between the contribution of our prompt design) with them and other related papers like [4]. Specifically, their prompt representations are statically learned for all input videos, or conditionally learned from video contexts, which introduces extra network parameters and tends to over-fit to base categories (when applied to the open-vocabulary setting). In contrast, our prompt representations are dynamically learned w.r.t motion patterns of different subject-object pairs.
>
> **Corresponding Revisions:**   Sec. 1 (**Line 85**) and Sec. 2 (**Lines 112 ~ 115, Lines 118 ~ 120**)  in the revised paper.
>
> [4] Sauradip Nag, Xiatian Zhu, Yi-Zhe Song, and Tao Xiang. Zero-shot temporal action detection via vision-language prompting. In ECCV, 2022.
>
> $\quad$
>
> ## Q3: Unclear key ideas and unnecessary notations/equations.
> > ... now it's very hard to grasp the key ideas, too many notions, and unnecessary equations.
>
> **A3:** Thank you for your suggestions.
> **For key ideas,** at first, we would like to reclaim our key ideas and main contributions in this paper. Our key idea is to consider the *compositional* and *motion-related* characteristics of the newly introduced Open-VidVRD task. To this end, we propose a compositional and motion-based prompt tuning framework, in which 1) the compositional prompt representation models the prompt contexts for subject and object separately; 2) the motion-cue-based prompt groups capture the general characteristic of object tracklet pairs.
>
> To improve the clarification of the key ideas, we have modified the writing in the revised manuscript.
>
> **Corresponding Revisions:**   Sec. 1 (**Lines 76 ~ 78 & Lines 80 ~ 82**) in the revised paper.
>
> **For notations and equations,** we have tried to simplify the notions and equations in the revised paper for clearer presentation. Specifically,
>
> - In Sec 3.2 (**Lines 153 ~ 155**), we removed the notations of bounding box sequence ($\mathbf{b}_i$), the length of tracklet ($l_i$), and the RoI feature before temporal averaging ($\mathbf{f}_i^0$), since they are not used in the following paragraphs.
>
> - We simplified the notation of object visual embedding by removing the superscript. Specifically, we modified $\mathbf{v}^o_i$ to $\mathbf{v}_i$ (in Sec. 3.1 **Line 141** and **Eq. (5)**), and modified $\mathbf{v}^{o\prime}_i$ to $\mathbf{v}'$ (in Sec. 3.2 **Line 160, Eq.(3)** and **Eq.(5)**).
>
> - We simplified the notation of tracklet pair's visual embedding and RoI features, i.e., we modified $\mathbf{v}^p_{i,j}$ to $\mathbf{v}\_{i,j}$ (in Sec 3.3 **Line 210** and **Eq.(9)**), and modified $\mathbf{f}^p_{i,j}$ to $\mathbf{f}_{i,j}$ (in Sec 3.3 **Line 226** and **Eq.(11)**).

---

> > ### Author Response · Authors · 2022-11-16
> > **Response to Reviewer QLPW (2/2)**
> >
> > ## Q4: Concern about the reproducibility.
> > > Reproducibility: the writing is not ideal, which makes it a bit difficult to reproduce, unless the author will release the code.
> >
> > **A4:** We had provided the code of models and train/eval scripts in the supplementary materials, which include both the RePro model and its variants for ablation studies. Please kindly take a look. We promise to make the whole project open upon the acceptance of the paper.

---

### Author Response · Authors · 2022-11-16
**General Response to All Reviewers**

We thank all reviewers for recognizing our paper well-written (Reviewers Fty6, Nj5q, K5cm), easy to follow (Reviewers Nj5q, K5cm), with novel ideas/frameworks/settings (Reviewers Fty6, Nj5q, K5cm), with rich experiments (Reviewers Fty6) and convincing results (Reviewer QLPW). We appreciate their suggestions and comments, and carefully revise our paper accordingly. Our major revisions include 5 aspects as follows:


1. In the Introduction section,  we improved the clarification of our key ideas (Lines 76 ~ 78, Lines 80 ~ 82, Line 85).
2. In the Related Work section,  we improved the writing to clarify the novelty of our Open-VidVRD setting compared to existing works with zero-shot setting (Lines 102 ~ 111).  We also included some references from ECCV which are related to our work (Lines 112 ~ 115, Lines 118 ~ 120).
3. In the Method section, we updated Figure 3 and its caption for a clearer presentation. We also simplified the notations and equations.
4. In the Experiments section, we improved the clarification of our motivations behind the evaluation and ablations (Lines 272 ~  283, Lines 292 ~ 299, Lines 302 ~ 306, and Lines 308 ~ 324). We also updated the new results of our RePro in conventional VidVRD setting trained on both base and novel category samples (Table 2).
5. In the Appendix:
   - We moved the detailed hyper-parameters settings into Sec. A.3.
   - We added the analysis of performance improvement in different predicate groups in Sec. A.4.
   - We added the potential improvements of the motion pattern design (as the future work) in Sec. A.5.
   - We added the detailed experiment settings of the compared SOTA methods in Sec. A.6.

Please note that we colorized the revisions in the new version of the paper.

---

### Author Response · Authors · 2022-12-12
**Thanks to Reviewers and AC**

Dear Reviewers and Area Chairs,

We sincerely appreciate your efforts in reviewing our paper, and your constructive comments. We have responded all the comments, revised the manuscript accordingly, and provided additional experiment results as requested.

We are wondering if there are any unsolved questions or concerns, and we are happy to answer them.

Best regards,

Paper4266 Authors.

---

### Decision · Program_Chairs · 2023-01-20

**Decision:**

Accept: poster

**Justification For Why Not Higher Score:**

Although the rebuttal have successfully addressed reviewers' initial concerns, the reviewers did not find the revision and the rebuttal particularly strong enough to warrant higher scores. As a result, although they all provided explicit endorsement for acceptance, they kept their initial ratings (6). This meta reviewer has carefully read the paper, the reviews, the rebuttal, and the discussion threads, and did not find strong reasons to upvote beyond the reviewers' suggestions.

**Justification For Why Not Lower Score:**

The paper has enough merits to appear at the conference (see above).

**Metareview: Summary, Strengths And Weaknesses:**

All four reviewers agreed that the paper makes good contributions with 1) a novel and challenging task of open-vocabulary visual relation detection in videos, 2) a novel prompt learning scheme that leverages compositional and motion-related priors, and 3) strong empirical results supporting the efficacy of the proposed approach.

The reviewers raised some clarity issues, which were well addressed during the rebuttal. They also expressed some concerns about experiments and provided good suggestions on additional experiments to further improve the quality of the paper. The authors provided additional results in the rebuttal which addressed the initial concerns.

Overall, there is a consensus that this paper has enough contributions to appear at the conference. The proposed task is novel and challenging, the proposed approach is reasonable, and experimental results are convincing to support their claims. Given this, we are happy to recommend acceptance.

**Note From Pc:**

if the above contains the word "oral" or "spotlight" please see: "oral" presentation means -> notable-top-5% and "spotlight" means -> notable-top-25%. As stated in our emails, we are disassociating presentation type from AC recommendations